# A Proteogenomic Approach to Unravel New Proteins Encoded in the *Leishmania donovani* (HU3) Genome

**DOI:** 10.3390/genes15060775

**Published:** 2024-06-13

**Authors:** Javier Adán-Jiménez, Alejandro Sánchez-Salvador, Esperanza Morato, Jose Carlos Solana, Begoña Aguado, Jose M. Requena

**Affiliations:** 1Centro de Biología Molecular Severo Ochoa (CSIC-UAM), Departamento de Biología Molecular, Instituto Universitario de Biología Molecular (IUBM), Universidad Autónoma de Madrid, 28049 Madrid, Spain; javier.adan@cbm.csic.es (J.A.-J.); alejandro.sanchez@cbm.csic.es (A.S.-S.); emorato@cbm.csic.es (E.M.); jcsolana@cbm.csic.es (J.C.S.); 2Centro de Investigación Biomédica en Red de Enfermedades Infecciosas, Instituto de Salud Carlos III, 28029 Madrid, Spain

**Keywords:** *Leishmania donovani*, experimental proteome, post-translational modifications (PTMs), proteogenomics, mass spectrometry

## Abstract

The high-throughput proteomics data generated by increasingly more sensible mass spectrometers greatly contribute to our better understanding of molecular and cellular mechanisms operating in live beings. Nevertheless, proteomics analyses are based on accurate genomic and protein annotations, and some information may be lost if these resources are incomplete. Here, we show that most proteomics data may be recovered by interconnecting genomics and proteomics approaches (i.e., following a proteogenomic strategy), resulting, in turn, in an improvement of gene/protein models. In this study, we generated proteomics data from *Leishmania donovani* (HU3 strain) promastigotes that allowed us to detect 1908 proteins in this developmental stage on the basis of the currently annotated proteins available in public databases. However, when the proteomics data were searched against all possible open reading frames existing in the *L. donovani* genome, twenty new protein-coding genes could be annotated. Additionally, 43 previously annotated proteins were extended at their N-terminal ends to accommodate peptides detected in the proteomics data. Also, different post-translational modifications (phosphorylation, acetylation, methylation, among others) were found to occur in a large number of *Leishmania* proteins. Finally, a detailed comparative analysis of the *L. donovani* and *Leishmania major* experimental proteomes served to illustrate how inaccurate conclusions can be raised if proteomes are compared solely on the basis of the listed proteins identified in each proteome. Finally, we have created data entries (based on freely available repositories) to provide and maintain updated gene/protein models. Raw data are available via ProteomeXchange with the identifier PXD051920.

## 1. Introduction

*Leishmania* is a protozoan parasite belonging to the order Trypanosomatida, and a causative agent of leishmaniasis in both humans and canids. The disease has a worldwide distribution and more than one billion people live at risk of infection [1]. Despite its high incidence, no acceptable vaccine for humans exists [2] and treatment relies on chemotherapy, but, currently, the drug’s arsenal is limited [3]. Moreover, global climate alterations are contributing to the spread of the *Leishmania*-transmitting vectors, the phlebotomine sand flies and, consequently, to increase the number of affected persons [4]. Therefore, it is urgent to develop new strategies to combat this parasitosis, and detailed knowledge of the molecular and cellular biology of this parasite will offer new avenues for struggling against it [5].

Around twenty *Leishmania* species have been described as human pathogens; although they are morphologically very similar, substantially different pathological outcomes can result after infection with the different species [6]. In humans, three main clinical manifestations of leishmaniasis occur: visceral leishmaniasis (VL, or kala-azar), cutaneous leishmaniasis (CL) and mucocutaneous leishmaniasis (MCL). The deadliest form is VL, caused by *L. donovani* (endemic in India and the Northeast of Africa) and *Leishmania infantum* (distributed in countries around the Mediterranean basin, North Africa and Latin America). The outcome of *Leishmania* infections is determined by a combination of host immunological status and pathogen virulence factors [7].

In the last decade, impressive methodological advances in molecular analytical techniques have occurred, allowing gathering information on the vast majority of cellular constituents (genes, transcripts, proteins and metabolites) of a whole cell/organism by a single experiment (Omics technologies). These omics approaches (genomics, transcriptomics, proteomics and metabolomics, among others) are being used for studying the different *Leishmania* species in order to understand the molecular biology of this parasite and the virulence factors responsible for the distinct pathological outcomes caused by the different species [5,8]. In particular, the determination of the protein compendium (proteome) being expressed in the different *Leishmania* species represents a quite valuable approach to directly depict predominant metabolic processes and virulence factors that may show some degree of species-specificity. Although relevant proteomics studies based on protein separation by two-dimensional gel electrophoresis allowed us to show global differences in proteomes between species and developmental stages (reviewed in [8,9]), the number of identified proteins was relatively low regarding the number of predicted genes existing in the *Leishmania* genome. Nevertheless, the high sensitivity of new mass spectrometers, together with improved bioinformatics tools for peptide spectra assignation, have led to the identification of large numbers of proteins in complex samples without accomplishing the cumbersome biochemical fractionation steps. Hence, in recent proteomics studies, thousands of different *Leishmania* proteins were identified. For instance, 1764 different proteins were identified in the *Leishmania mexicana* intracellular (amastigote) form [10] and 2711 in the extracellular (promastigote) one [11], 2428 proteins were identified in *L. donovani* amastigotes [12], 1212 proteins were identified in *Leishmania tropica* promastigotes [13], 3883 different proteins were identified after subcellular fractionation of *L. donovani* promastigotes [14], 1713 different proteins were identified during the *L. donovani* promastigote-to-amastigote axenic differentiation follow-up [15], 2352 different proteins were identified in *L. infantum* promastigotes [16], over 6500 different proteins were identified in *L. major* promastigotes [17] and numbers above 6700 each for the different proteins identified in promastigotes of three species of the Viannia subgenus, *Leishmania braziliensis*, *Leishmania panamensis* and *Leishmnia guyanensis* [18]. The protein identification in most of these proteomic studies was made using the available predicted proteomes in dedicated databases (TriTryDB, NCBI/ENA and UniProt), but proteogenomics approaches were not usually performed.

Since mass spectral data identification engines rely on already existing protein databases, complete and well-annotated genomes are essential resources for accurate and detailed analyses of whole-cell-based studies [19]. However, genomic annotations do not conclude after determining the genome sequence and performing the bioinformatics predictions on gene content; improvements in genomic annotations are continuously incorporated on the basis of experimental data. Here, we conducted a proteomic study of *L. donovani* (HU3 strain) promastigotes with two main objectives. On the one hand, we analyzed the experimental proteome of the promastigote stage and compared it with those from other *Leishmania* species to identify species-specific proteins that might be virulent factors responsible for the severe pathologies that the infection by this species produces. On the other hand, we used the proteomic data for improving the annotations of the *L. donovani* (HU3 strain) genome; as a result, new protein-coding genes have been uncovered and coding sequences extended for some previously annotated genes.

## 2. Materials and Methods

### 2.1. Parasite Culture and Preparations of Samples

Promastigotes of *L. donovani* of the HU3 strain (WHO code: MHOM/ET/67/HU3) were grown at 26 °C in Roswell Park Memorial Institute (RPMI) medium supplemented with 10% of heat-inactivated fetal bovine serum (FBS), hemin (10 μg/mL) and an antibiotic mix (streptomycin 10 μg/mL and penicillin 105 U/mL). Cultures (50 mL) were started at 5 × 10^5^ cells/mL and the parasites were harvested in the middle logarithmic growth phase (10^7^ promastigotes/mL). After washing twice with phosphate buffer saline (PBS), the pellets (5 × 10^8^ cells) were processed following two different procedures (see below).

### 2.2. Preparation of Protein Extract in STRAP Buffer and Digestion in Column (S-Trap Mini)

A pellet consisting of 5 × 10^8^ promastigotes (see above) was suspended in 300 µL of S-TRAP buffer: 5% SDS, 7 M urea, 2 M thiourea and 50 mM triethylammonium bicarbonate (TEAB) pH 8.5. The sample was sonicated by the UP100H Ultrasonic Processor (Hielscher, Teltow, Germany) applying a total of 20 pulses (4 cycles of 5 pulses) at 100% amplitude. The tubes were cooled on ice after each cycle. Then, the sample was centrifuged at 13,000× *g* for 10 min at 4 °C, and the supernatant was analyzed by SDS-PAGE. After Coomassie blue staining, the protein concentration was estimated to be around 7 mg/mL.

The sample (150 µg) was digested using the S-TRAP: Rapid Universal MS Sample Prep Columns (PROTIFI, Fairport, NY, USA) following the supplier’s instructions with minor modifications. Briefly, the protein extracts (adjusted to 50 μL with S-TRAP buffer) were reduced and alkylated (disulfide bonds from cysteinyl residues were reduced with 10 mM DTT for 1 h at 37 °C, and then thiol groups were alkylated with 10 mM iodoacetamide for 1 h at room temperature in darkness). Then, 0.1 volume of phosphoric acid was added to a final concentration of 1.1%; this step is essential to completely denature proteins and trap them in the S-Trap column efficiently. At this point, the pH should be ≤ 1. Afterward, the sample was diluted with 7 volumes of a mixture consisting (in a 1:7 ratio) of S-trap Binding Buffer and a solution of 90% methanol and 100 mM TEAB. The sample was digested in a column with sequencing grade trypsin (Promega, Madison, WI, USA) with a 1:25 ratio (protease: protein) and incubated for 2 h at 47 °C in a ThermoMixer. The column was eluted by the addition of 80 µL of elution buffer (80% acetonitrile (CAN), 0.2% formic acid) and centrifuged for 1 min at 4000× *g*. The process was repeated and the two eluates were pooled and dried down in a Speedvac device. The digested peptides (30 μg) were desalted by loading them onto OMIX Pipette tips C18 (Agilent Technologies, Santa Clara, CA, USA) before the mass spectrometric analysis (see below).

### 2.3. In-Gel Digestion

Thirty µL of the S-TRAP buffer-cellular extracts (see above) were mixed with 50 µL of Laemmli buffer, and then 7.5 µL were applied onto 1.2 cm wide wells of a conventional SDS-PAGE gel (0.75 mm thick, 4% polyacrylamide in the stacking gel and 10% polyacrylamide in the resolving one). Electrophoresis was stopped as soon as the front entered 3 mm into the resolving gel. The unseparated protein bands were visualized by Coomassie staining, excised from the gel, which was cut into cubes (2 × 2 mm), and placed in 0.5 mL microcentrifuge tubes, as described elsewhere [16]. The gel pieces were destained in ACN/water (1:1) solution, and then reduced, alkylated (disulfide bonds from cysteinyl residues were reduced with 10 mM DTT for 1 h at 56 °C and then thiol groups were alkylated with 10 mM iodoacetamide for 30 min at room temperature in darkness) and digested either with sequencing grade trypsin (Promega, Madison, WI, USA) or chymotrypsin (Roche, Mannheim, Germany), as described by Shevchenko et al. [20], with minor modifications. The gel pieces were shrunk by adding an excess of ACN to remove water. Finally, after pipetting out the ACN solution, gel pieces were dried in a Speedvac. The dried gel pieces were re-swollen in 100 mM Tris-HCl pH 8, 10 mM CaCl_2_ with 60 ng/µL trypsin (or chymotrypsin) at 5:1 protein/enzyme (*w*/*w*) ratio. The tubes were kept in ice for 2 h and incubated at 37 °C (trypsin) or 25 °C (chymotrypsin) for 12 h. Digestion was stopped by the addition of 1% TFA. Whole supernatants were dried down and then desalted onto OMIX Pipette tips C18 (Agilent Technologies) before the mass spectrometric analysis.

### 2.4. Reverse Phase-Liquid Chromatography (RP-LC)-MS/MS Analysis (Dynamic Exclusion Mode)

After drying the enzymatically digested protein samples (see Section 2.3), these were suspended in 10 µL of 0.1% formic acid to be analyzed by RP-LC-MS/MS in an Easy-nLC 1200 system coupled to an ion trap LTQ-Orbitrap Velos Pro hybrid mass spectrometer (Thermo Scientific, Waltham, MA, USA). The peptides were concentrated (online) by reverse phase chromatography using a 0.1 mm × 20 mm C18 RP precolumn (Thermo Scientific) and then separated using a 0.075 mm × 250 mm bioZen 2.6 µm Peptide XB-C18 RP column (Phenomenex, Torrance, CA, USA) operating at 0.25 μL/min. Peptides were eluted using a 180 min dual gradient. The gradient profile was set as follows: 5–25% solvent B for 135 min, 25–40% solvent B for 45 min, 40–100% solvent B for 2 min and 100% solvent B for 18 min. Solvent A consisted of 0.1% formic acid in water, and solvent B was a mixture of 0.1% formic acid and 80% acetonitrile in water. Electrospray ionization (ESI) was carried out using a nano-bore emitter stainless steel ID 30 µm (Proxeon, Odense, Denmark) interface at 2.1 kV spray voltage with S-Lens of 60%. The Orbitrap resolution was set at 30,000. Peptides were detected in survey scans from 400 to 1600 amu (1 µscan), followed by twenty data-dependent MS/MS scans (Top 20) using an isolation width of 2 u (in mass-to-charge ratio units), a normalized collision energy of 35% and a dynamic exclusion that was applied during 60 s periods. Charge-state screening was enabled to reject unassigned and singly charged protonated ions.

### 2.5. Data Analysis

Peptide identification from raw data was carried out using the PEAKS Studio XPro search engine (Bioinformatics Solutions Inc., Waterloo, ON, Canada) [21]. Searches were performed against two databases: (i) current *L. donovani* (HU3 strain) proteome available at UniProt (ID: UP000601710; [22]), and (ii) a database consisting of all possible open reading frames (ORF) coding for protein sequences of ≥20 amino acids existing in any of the six frames of the *L. donovani* (HU3 strain) genome (this database, henceforth, is named LdHU3-all-ORFs). This database is publicly available in the Mendeley data repository through the link: https://data.mendeley.com/datasets/6b54424fgs/1 (accessed on 13 May 2024). As controls, mass spectra were searched against the corresponding decoy databases (decoy fusion database). The following constraints were used for the searches: tryptic cleavage after Arg and Lys (semispecific) or chymotryptic cleavage after Tyr, Trip, Phe and Leu, up to two missed cleavage sites, and tolerances of 20 ppm for precursor ions and 0.6 Da for MS/MS fragment ions; also, the searches were performed allowing optional Met oxidation and Cys carbamidomethylation. False discovery rates (FDR) for peptide spectrum matches (PSM) and for protein were limited to 0.01. Only those proteins with at least two unique peptides discovered from LC/MS/MS analyses were considered reliably identified.

### 2.6. Data Availability

The mass spectrometry proteomics data have been deposited to the ProteomeXchange Consortium via the PRIDE [23] partner repository with the dataset identifier PXD051920 and 10.6019/PXD051920.

Improved sequence annotations are available as Mendeley datasets (https://data.mendeley.com/, (accessed on 13 May 2024)), and they can be searched by using the gene ID and/or functional gene annotation.

## 3. Results

### 3.1. L. donovani (HU3) Experimental Proteome Determined by Protein Identification from LC−MS/MS Peptide Spectra

Total protein extracts from *L. donovani* (HU3) promastigotes were digested by trypsin or chymotrypsin following two methodological procedures (see Methods for additional details). Afterward, the peptide mixtures were analyzed by mass spectrometry and the predicted masses were searched against a database consisting of the annotated proteome for this species (UniProt ID: UP000601710). In this survey, only proteins identified by two or more peptides were considered for further analyses. Hence, as a result, 1908 proteins were considered to be trustily identified (see Appendix A for the complete list).

Recently, Prof. Beverley’s group reported the identification of 6208 different proteins in *L. major* promastigotes [17]. Certainly, the proteome coverage detected by these authors was significantly larger than that attained by us in this work (1908 proteins). If we consider all the proteins identified by one or more peptides (the criteria used by Polanco et al.), the number of identified proteins in our assay would increase to 2648. Considering the deep coverage of the *L. major* promastigote proteome attained by Polanco et al. [17], we expected that all of the proteins identified in the *L. donovani* promastigote experimental proteome would have orthologues among the *L. major* identified proteins; otherwise, we would be evidencing *L. donovani* species-specific proteins. After crossing both experimental proteomes (Figure 1), the results indicated that the *L. donovani* experimental proteome contained 239 proteins that were not presumably identified in the *L. major* experimental proteome. This finding was clearly unexpected, as the number of species-specific genes among the Old World *Leishmania* species *L. infantum* (a close relative to *L. donovani*) and *L. major* was determined to be only 27 [24]. To decipher the meaning of these results, we separated the 239 proteins into two groups: those having annotated orthologous genes in the *L. major* (Friedlin) genome (the number was 190) and those without annotated orthologues (amounting to 49). Hence, a detailed analysis of the *L. donovani* proteins without apparent orthologue in *L. major* showed that most of these proteins are encoded by repeated genes (β-tubulin, HSP70, histones and ribosomal proteins, among others). In fact, 44 out of the 49 entries belong to this category and, therefore, these proteins do not represent species-specific genes as they were also identified by Polanco and co-workers in the *L. major* experimental proteome [17]. Another four proteins (LDHU3_23.2440, LDHU3_26.1830, LDHU3_29.0450 and LDHU3_34.1980) lacked annotated orthologues in the *L. major* Friedlin (LmjF) reference genome [25]. However, three out of these four proteins have annotated orthologues (LMJFC_230027200, LMJFC_290008800 and LMJFC_340020400) in a more recent genome assembly (LMJFC) generated for the same *L. major* Friedlin strain [26]. Moreover, the orthologue for LDHU3_26.1830, although not currently annotated in the LMJFC genome, was found encoded in transcript LMJFC_260021700, previously annotated as non-coding RNA. All the mentioned gene entries are available in the TriTrypDB database (*L. major* Friedlin 2021) and the information regarding the newly annotated gene LMJFC_260021700 is now available as a Mendeley data entry (https://data.mendeley.com/datasets/8d8wt3mgty/1, accessed on 13 May 2024).

In sum, among this group of 49 entries, only protein LDHU3_02.0870 might be an *L. donovani*-specific protein. Previously, this entry was labeled as a pseudogene, but now the proteomic data showed the existence of three peptides (one unique) mapping to an ORF encoding a polypeptide of 155 amino acids in length. Remarkably, the sequence 2–138 of this protein contains a motif typical of the peptidase M3A/M3B family (InterPro motif: IPR045090). Therefore, this entry should be re-annotated as a protein-coding protein in the *L. donovani* (HU3) genome, and its sequence is now available in the Mendeley data repository (https://data.mendeley.com/datasets/svcr7j8p4y/1, accessed on 13 May 2024).

On the other hand, after crossing both experimental proteomes, 190 of the *L. donovani* proteins, having annotated orthologues in the LmjF genome, were filtered out as non-detected in the *L. major* experimental proteome determined by Polanco and coworkers [17]. However, a detailed analysis showed that most of the missing proteins are paralogous copies of other listed proteins in the *L. major* proteome. Thus, we realized that these authors only listed one paralogous for each protein group; nevertheless, this does not mean that the other paralogous proteins are not expressed. Thus, 183 out of the 190 *L. donovani* proteins comprising this category were considered to have been identified in both experimental proteomes. For another five cases, the annotated orthologues pair (*L. donovani* vs. *L. major*) were not true orthologues. These wrong-matched orthologous pairs are LDHU3_19.0350 and LmjF.19.0305, LDHU3_24.0910 and LmjF.24.0765, LDHU3_30.1830 and LmjF.30.1380, LDHU3_35.3520 and LmjF.35.2725, and LDHU3_36.3310 and LmjF.36.2350. In fact, the true orthologues were not annotated in the *L. major* Friedlin reference genome (LmjF); however, three of them were recently annotated in the genome for this strain, which was re-assembled in 2021 [26], and they correspond to LMJFC_190008900, LMJFC_240013800 and LMJFC_300020900 entries. In addition, orthologues to the other *L. donovani* proteins (LDHU3_35.3520 and LDHU3_36.3310) have been annotated in another *L. major* strain (LMJLV39_350034000 and LMJLV39_360031000, respectively). Therefore, we cannot consider that any of these five proteins are not expressed in the *L. major* promastigotes but they were not identified because of the use of an incomplete *L. major* genome assembly (LmjF). In sum, only two *L. donovani* proteins (LDHU3_33.1690 and LDHU3_34.5270) with bona fide *L. major* orthologues (LmjF.33.1035 and LmjF.34.3330) were detected in the *L. donovani* proteome reported here but are absent in the *L. major* proteome determined by Polanco and coworkers [17]. Protein LDHU3_33.1690 (orthologue: LmjF.33.1035, hypothetical protein) was identified by four unique peptides and protein LDHU3_34.5270 (LmjF.34.3330, cytochrome p450-like protein) by three unique peptides. Further experimental analyses would be required to determine whether these proteins are specifically expressed in *L. donovani* promastigotes but not in the *L. major* ones.

In summary, only 1 out of the 49 initially postulated as species-specific *L. donovani* proteins identified in the proteome (Figure 1) might be certain. And only 2 out of the 190 *L. donovani* proteins having *L. major* orthologues (apparently not identified in the *L. major* proteome) remained after an in-depth analysis as possible stage-specific differentially expressed in *L. donovani* promastigotes. These analyses have evidenced that proteome identification would benefit from having well-annotated genomes and curated databases; otherwise, false conclusions may arise with ease.

### 3.2. Annotation of New Protein-Coding Genes in the L. donovani (HU3) Genome

Proteomic data were also analyzed through a proteogenomic approach, in which experimental mass spectra were matched against a theoretical protein database created by translating into protein sequences every possible open reading frame (ORF) existing in the *L. donovani* (HU3) genome. Figure 2 illustrates the experimental and bioinformatics procedures. When the MS/MS spectra were analyzed using the current proteome annotated for the *L. donovani* (HU3 strain) available in the UniProt database, 18,016 peptides were identified. Interestingly, however, when the analysis was repeated using the database with all possible ORFs (LdHU3-all-ORFs database), the number of identified peptides was 20,377 peptides. Consequently, a large fraction of peptides (i.e., 2361) would correspond to genomic regions previously considered as non-coding. A detailed analysis of the location of these new peptides allowed us to annotate coding sequences in 20 transcripts (Table 1), which were previously annotated as non-coding RNAs (ncRNA) or pseudogenes [22]. Some of those genes correspond to conserved genes, already annotated in other *Leishmania* species. Hence, these omissions may be attributable to errors during the automatic annotation of the genome. Nevertheless, four protein-coding genes were not previously annotated in any of the reference *Leishmania* genomes; these are LDHU3_22.1300, LDHU3_30.5010, LDHU3_32.4600 and LDHU3_36.7950. However, in a previous study, in which ribosome-protected mRNA fragments (Ribo-Seq) were analyzed, it was already pointed out that these might be protein-coding genes [27]. Additionally, the proteins encoded by the orthologs to LDHU3_32.4600 in *L. infantum* (LINF_320041950) and to *L. major* Friedlin (LMJFC_320046900) were evidenced in previous proteomic studies [16,28]. In Figure 3, as an example, we show the experimental data supporting that gene LDHU3_22.1300 should be categorized as a protein-coding gene. Although the polypeptide encoded by this gene is small (66 amino acids), three peptides were found to fit well with the experimental mass spectra and all peptides were unique for this sequence (Figure 3A). Moreover, it was possible to find this ORF in the genomic sequences of other *Leishmania* species (Figure 3B). Hence, a conserved ORF was found in the *L. major* (Friedlin) transcript LMJFC_220016800_t1, which is 867 nucleotides in length and is currently annotated as ncRNA_gene (https://tritrypdb.org, accessed on 13 May 2024). In the *L. mexicana* reference genome (MHOM/GT/2001/U1103), annotated transcripts are not available, but the ORF could be located at chromosome 22 (LmxM.22: 389689-389830). Finally, the gene coding for the orthologue protein (LINF_220015750) was previously identified in *L. infantum* (JPCM5) following also a proteogenomic strategy [16]. As shown in Figure 3B, the protein sequence is well conserved among these *Leishmania* species.

### 3.3. Re-Annotation of Gene Coding Sequences (CDS) to Accommodate Peptides Identified by the Proteomics Data

A significant number of mass spectra were found in a database consisting of all theoretical ORFs existing in the *L. donovani* (HU3) genome (LdHU3-all-ORFs database), but absent from the currently annotated proteome (https://www.uniprot.org/). These allowed us to annotate new protein-coding genes (see Section 3.2), but also to extend the CDS at its 5′ end for 43 genes (see Appendix A). Figure 4 illustrates the rationale leading to the modification of the CDS for gene LDHU3_01.0360 (the first listed in the Appendix A). The currently annotated CDS encodes for a polypeptide of 307 amino acids in length (the protein is annotated as a poly(A) export protein); however, we found two additional peptides mapping on an extended ORF (coding for a polypeptide of 339 amino acids), suggesting that the current CDS was erroneously annotated. Remarkably, similar shortened CDS were annotated for the orthologous genes in other species: LMJFC_010008400 in *L. major* Friedlin, LINF_010008200 in *L. infantum* JPCM5 and LmxM.01.0320 in *L. mexicana* U1103 (Figure 4). Nevertheless, in previous work, based also on proteomics data, the CDS for the *L. infantum* LINF_010008200 gene was extended [16]. However, general sequence repositories maintain the initial annotation; to overcome this difficulty, we opted to create Mendeley datasets in which curated sequences may be immediately incorporated (and, consequently, downloaded by anyone interested in them). In this case, the actual sequences for LINF_010008200 gene/protein may be downloaded from the Mendeley data repository (https://data.mendeley.com/) by searching for this ID (http://dx.doi.org/10.17632/rz69zd9ftf.1, accessed on 13 May 2024).

Among the 43 proteins that were extended at their N-terminal, we briefly comment on some noticeable findings. Protein LDHU3_04.0600 corresponds to an adenylosuccinate lyase, and in the extended sequence, it was found an acetylation in the serine follows the initial methionine, suggesting that the enzyme might be regulated by this post-translational modification. In the extended sequences of proteins LDHU3_11.1160 and LDHU3_20.1390 (both coding for proteins of unknown function, but conserved among trypanosomatids), and LDHU3_30.4150 (RNA-binding protein 42) were also mapped N-terminal peptides with an acetylated residue at positions 1 or 2. An N-terminal extension of 130 amino acids was incorporated into protein LDHU3_15.0170 (coding for an ATP-dependent RNA helicase, which has been involved in ribosome assembly in *L. major* [30]). Similarly, the sequence of protein LDHU3_17.1500 (coding for NatC N(α)-terminal acetyltransferase) was extended from 462 amino acids in the current annotation (TriTrypDB) to 741 amino acids after the curation made in this work. It should be noted that all the protein sequences extended at their N-terminal end in this study were found to be misannotated in a recent article in which CDS annotations were curated in light of Ribo-seq data [27].

### 3.4. Identification of Two InDels in the Assembled L. dononani Genome Based on Proteomics Data

In the analysis of identified peptides fitting in the database consisting of all theoretical ORFs but absent from currently annotated proteins, we found a couple of cases in which proteomics data pointed to a possible punctual error in the *L. donovani* (HU3) genome sequence. As an example, Figure 5 illustrates how a sequence point error in gene LdHU3_27.3580 was uncovered, allowing curation of the nucleotide sequence of this gene. From the mass spectra, many peptides could be mapped to the current annotated protein for gene LdHU3_27.3580 (Figure 5A, upper panel). However, a search of peptides mapping only on the LdHU3-all-ORFs database, pointed to the existence of an ORF, overlapping with the LdHU3_27.3580 CDS (Figure 5A, bottom panel). This prompted us to analyze the alignment of the Illumina DNA-seq reads on the genomic region in which the gene LdHU3_27.3580 is located. As shown in Figure 5B, at both sides of chromosome 27 nucleotide position 1.099.456, most of the aligned reads were marked with an “I” (insertion), denoting that a G nucleotide exists in the reads that are missing from the assembled sequence. The reason that this nucleotide was not added to the final assembly by the bioinformatics tools is not clear, but there is no doubt about its existence. In fact, when this nucleotide is inserted, the CDS is extended (Figure 5C) and the encoded protein (LdHU3c in the figure) is now 100% identical to the orthologous JDP2 protein in *L. infantum* (LINF_270032200). We have created a Mendeley dataset with the curated sequences (gene, CDS and protein) for LdHU3_27.3580 (https://data.mendeley.com/datasets/zrty4xzhz3/1, accessed on 13 May 2024). The second case affected the gene LDHU3_31.1660; the proteomics data alerted on the existence of peptides mapped to two overlapping ORFs and the manual inspection of the Illumina reads pointed that a G nucleotide should be inserted in the *L. donovani* (HU3) genome sequence. Thus, after curation of the sequence, the CDS was extended and found that the encoded protein is identical in sequence and length to that encoded in the *L. infantum* LINF_310015900 gene (coding for a protein of unknown function, but conserved among trypanosomatids). Also, we have created a Mendeley dataset with the curated sequences (gene, CDS and protein) for LDHU3_31.1660 (https://data.mendeley.com/datasets/tbpg5ztvnw/1, accessed on 13 May 2024).

### 3.5. Identification of Post-Translational Modifications (PTMs) in L. donovani Proteins

Post-translational modifications (PTMs) in proteins are critical for regulating their activity, subcellular localization, physicochemical properties, lifespan and functional interactions with other molecules. This layout of regulation is especially relevant for *Leishmania* (and related trypanosomatids) in which gene regulation does not operate at the transcriptional level [32]. The advances in proteomics techniques and bioinformatics tools make the identification of particular modified residues in proteins a feasible task [9]. PTMs occurring in a protein sequence result in a characteristic mass shift that is readily measured by MS. In this work, we have identified a large number of *L. donovani* proteins with physiological post-translational modifications, and the results are commented on briefly in the context of particular PTMs.

Site-specific phosphorylation of proteins is the most studied PTM because of its relevance in controlling cellular signaling networks. However, this PTM is usually transient since protein kinases and phosphatases compete in a dynamic way to add (or remove) a phosphate group to (or from) a specific amino acid residue. Hence, the number of detected phosphosites in proteins is usually lower than the real one; alternatively, specific inhibitors of phosphatases are used and/or procedures to obtain the enrichment of phosphopeptides are followed. In our proteomics data, only 23 phosphoproteins were identified (listed in the Appendix A). Some of them are proteins of known functional relevance: LDHU3_05.0130 (encoding for a phosphoprotein phosphatase located at the flagellar pocket [33]), LDHU3_18.0340 (coding for the glycogen synthase kinase 3 (GSK3)), LDHU3_26.2570 (encoding for the nucleolar protein 86, whose orthologue in *Trypanosoma brucei* was found to be essential for mitotic progression [34], LDHU3_32.3890 (encoding for a nucleoside diphosphate kinase b (NDPK1) that would be playing also a role in parasite infectivity [35]) and LDHU3_36.8060 (KHARON1, a protein essential for executing cytokinesis in *Leishmania* [36]). Remarkably, most of the phosphoproteins identified in this study were also identified to be phosphorylated in *L. infantum* orthologues [16]. The detected phosphorylations mostly occurred on Ser (S), and are less frequent on residues Tyr (Y) and Thr (T).

Acetylation of proteins, mainly at its N-terminal end, is one of the most widespread protein modifications, particularly in eukaryotic organisms [37]. Accordingly, we have identified peptides with acetylated residues for 118 out of the 1909 proteins identified in this work (listed in the Appendix A). In 30 proteins, the acetylated residue was the initial methionine, and in 76 the acetylation occurred at the second amino acid (48 in Ser, 19 in Ala and 9 in Thr). Acetylation of the initial methionine is accomplished by N-terminal acetyltransferases type B (NatB), whereas NatA catalyzes co-translational acetylation of proteins at N termini that have been processed by methionine aminopeptidases [38]. Although these acetyltransferases have not been characterized to date in *Leishmania*, the existence of such frequent PTM among the identified proteins represents strong evidence of the existence of these substrate-specific acetyltransferases in this parasite. In this regard, the identification of an acetylated methionine at position 5 of the amino acid sequence for protein LDHU3_10.1300 (a FKBP-type peptidyl-prolyl cis-trans isomerase) might indicate that this methionine might be the first translated methionine from the LDHU3_10.1300 transcript instead of the currently annotated one.

Reversible lysine acetylation has been implicated in a variety of cell-signaling processes by modulating protein–protein interactions. However, the study of this PTM is challenging due to its generally very low stoichiometry, ranging from 0.02% to 1% [39]. In our work, acetylated lysines were identified only in three proteins: LDHU3_15.1420, which is a tryparedoxin peroxidase whose kinetics parameters were characterized by Flohé and co-workers [40]; LDHU3_24.1020, which is a triosephosphate isomerase [41]; and LDHU3_33.1760 (an uncharacterized guanylate kinase).

Acetylation of either serine or threonine residues is often found at position 2 in co-translationally processed proteins at their N-terminal ends; this is accomplished by NatA-type acetyltransferases (see above). However, these PTMs were also observed in internally located residues of proteins [42]. In this regard, acetylation can compete with phosphorylation of the same residues (Ser or Thr), altering, in turn, the course of signaling pathways. From the proteomic data described here, we identified acetylated serine residues at internal positions in the following proteins: LDHU3_35.7050 (SAC3/GANP/THP3-like protein), LDHU3_36.6020 (paraflagellar rod component), LDHU3_36.6580 (protein of unknown function) and LDHU3_36.8540 (sucrase/ferredoxin-like family protein). Acetylation of internally located threonine residues was only identified in protein LDHU3_26.1960 (thimet oligopeptidase). Other acetylated residues located at internal positions were histidine 67 in protein LDHU3_10.1560 (PAB1-binding protein [43]), asparagine 45 in protein LDHU3_26.0790 (HSP10 chaperonin [44]) and methionine 548 in protein LDHU3_33.3610 (mitochondrial HSP75 [45]).

Post-translational modification of proteins by methylation has been explored mainly in histones, regarding its role in regulating chromatin compaction and gene expression [46]. However, in recent years, it has become clear that the incorporation of methyl groups at particular positions also is relevant to control protein function in other cellular compartments. In this regard, it is noticeable that among the 1909 *L. donovani* proteins identified in this study, 202 of them presented at least a methylated residue (listed in the Appendix A). Protein arginine methylation (aka, R-methylation) is a well-known PTM in mammals, in which a large family of protein arginine methyltransferases (PRMTs) have been characterized [47]. PRMTs regulate key cellular processes: transcription, RNA splicing, DNA repair, cell cycle and cell signaling networks. In *Leishmania*, five different PRMTs have been identified and characterized [48]. These PRMTs are particularly abundant at the promastigote stage, most of them have a cytoplasmic location and RNA-binding proteins (RBPs) were found to be predominant targets of these R-methylases [48]. In agreement with this observation, we identified several RBPs among the proteins having methylated residues (Table 2). Among the 202 proteins identified as methylated in *L. donovani*, R-methylation was found to be frequent, but also methylation of acidic residues (mainly glutamic (Glu, E) ones) was observed in a large number of proteins. These results are not unexpected as methylation at aspartic (Asp, D) and Glu residues have been found in both human and yeast cells [49], and also in *Leishmania* [50]. Other protein residues found to be methylated were lysine (K) and, less frequently, threonine (T) and serine (S). Methylation of these residues has been described as plausible PTM in proteins [51]. According to our data, α- and β-tubulin in *L. donovani* are highly methylated, and it is remarkable that methylated residues in β-tubulin are acidic ones, whereas α-tubulin methylations occur mainly at basic residues (R and K); this is suggestive of a possible role of methylation in modulating the ionic interactions between both tubulin subunits. Other highly methylated proteins detected in this study were: enolase (LDHU3_14.1580), glutamate dehydrogenase (LDHU3_15.1360), HSP60 (LDHU3_36.2780), cytoplasmic HSP70 (LDHU3_28.3970), mitochondrial HSP70 (LDHU3_30.3330), HSP70.4 (LDHU3_26.1510), HSP83/90 (LDHU3_33.0460) and elongation factor 2 (LDHU3_36.0280). Heat shock proteins (HSPs) play relevant roles in protein folding processes; in this regard, within the ‘protein folding’ category, a significant number of proteins were found to be modified by methylation (Table 2). Finally, other remarkable functional categories having a significant number of methylated proteins are oxidative stress, flagellar proteins and proteases (Table 2).

Other less frequent PTMs detected in this proteomic study were a formylation of Asp at position 181 (D181) of mitochondrial HSP70 (LDHU3_30.3330) and ubiquitination of the polyadenylate-binding protein 2 (LDHU3_35.5420) at lysine-527 (K527).

### 3.6. Active Curation of L. donovani (HU3) Gene Annotations

The *L. donovani* (HU3 strain) genome was assembled in 2019 by Camacho et al. [22], and the sequence and gene annotations were incorporated into ENA/GenBank, TriTryDB and UniProt databases. More recently, Sánchez-Salvador and coworkers [27] carried out an extensive curation on gene models based on sequencing of ribosome-protected mRNA fragments (Ribo-seq data). Consequently, a new annotation file was deposited at the ENA/GenBank repository (GCA_900635355.2). However, to date, this new information has not been incorporated into TriTryDB and UniProt repositories, which are essential resources for researchers working on the molecular biology of trypanosomatids. Moreover, gene/protein annotations are being continuously improved on the basis of new experimental data, as occurred in this study.

To fill this gap, we are incorporating the *L. donovani* gene models into two secure cloud-based repositories specialized in archiving structured data: Wikidata (https://www.wikidata.org/, accessed on 31 May 2024) and Mendeley data (https://data.mendeley.com/, accessed on 31 May 2024). Both repositories are publicly accessible and the stored data become immediately available after uploading. Users can explore both repositories, which are interlinked, simply by browsing these databases by gene IDs and/or functional annotations; moreover, for the Wikidata entries, users can contribute with annotations using the simple and intuitive interface that this repository provides. The sequences for genes, CDS and proteins are included in the Mendeley data entries, whereas Wikidata entries also include links to information available at the TriTrypDB and UniProt repositories. Also, the *L. donovani* gene/protein entries are linked to the Wikidata for the corresponding *L. infantum* (JPCM5) orthologues. The species *L. infantum* has been chosen as a reference for this project, and the Wikidata/Mendeley data entries for this species include also bibliographic information related to studies dealing with a given gene/protein belonging to any *Leishmania* species. However, these repositories are not substitutive of either general or dedicated repositories (i.e., TriTrypDB, UniProt and NCBI/ENA) that contain bioinformatics tools of enormous value for research activities. The goal is that Wikidata/Mendeley data efforts will maintain updated gene/protein annotations by adding as quickly as possible those experimental data that are being continuously generated by the *Leishmania* research community. Finally, the new annotations have to be incorporated into the general repositories.

## 4. Conclusions

A proteogenomics strategy combines proteomics data with genomic sequences (and sometimes also uses transcriptomic data) to enhance the identification of peptide spectra generated in proteomics analyses. In this strategy, a theoretical protein database is created from the genome sequence and used for peptide identification by matching mass spectra against a non-biased protein database. Hence, following a proteogenomic approach, this study allowed us to identify 20 novel protein-coding genes not previously annotated in the *L. donovani* genome [22]. In addition, it was possible to correct annotations of 43 gene models. This approach, previously used by other authors in the field [52], would be widely used to exploit the valuable data that large-scale mass spectrometry studies generate.

Additionally, in this study, we identified physiologically relevant post-translational modifications (phosphorylation, methylation and acetylation) in a large fraction of *Leishmania* proteins. In many organisms, these PTMs have been shown to be involved in regulating protein activity, stability and turnover rate as well as modulators in cellular signaling pathways. However, to date, there are few studies focused on *Leishmania* PTMs [16,50,53,54].

Another conclusion is that the usage of incomplete or non-updated databases may cause a loss of valuable proteomics data, precluding for instance the identification of relevant virulent factors whose characterization might be paramount to combat this parasite. Therefore, efforts should be made to curate current gene models, to gather experimental data and to make these improvements available in a quick and easy manner to other researchers working in the field.

The application of bioinformatics analyses to a well-established proteome will allow in silico identification of promising antigens, based on their antigenicity profile, to develop sensitive and specific serodiagnostic tools. Also, the identification of *Leishmania* proteins having major histocompatibility complex (MHC) class I- and/or II-restricted epitopes will help to develop protective vaccines for human use. On the other hand, bioinformatics-guided structural predictions and molecular docking analyses on the *Leishmania* proteome will accelerate the uncovering of novel therapeutics for the control of leishmaniasis.

## Figures and Tables

**Figure 1 genes-15-00775-f001:**
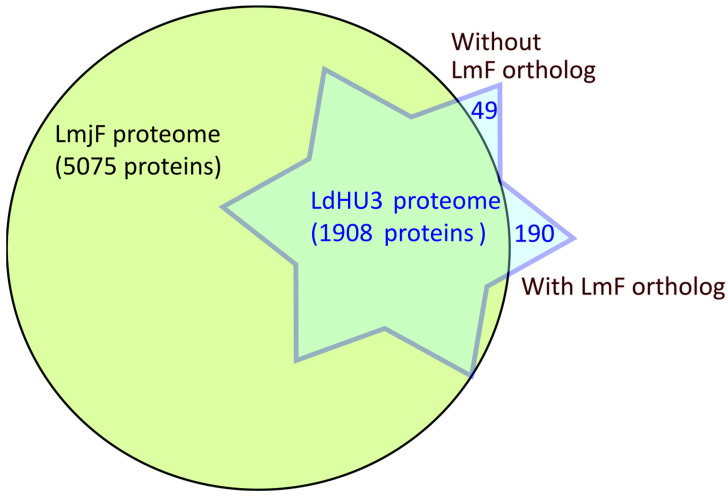
Venn plot showing the overlapping between proteins previously identified in *L. major* promastigotes and the proteins identified in *L. donovani* promastigotes in this study. The analysis was centered on the 5075 proteins identified in the wild-type *L. major* Friendly strain (LmjF, circle) by Polanco and coworkers [17]. Among the 1908 proteins identified in this study (LdHU3, star), 49 apparently lacked orthologs in *L. major* and 190 proteins seemed to be expressed exclusively in the *L. donovani* promastigotes.

**Figure 2 genes-15-00775-f002:**
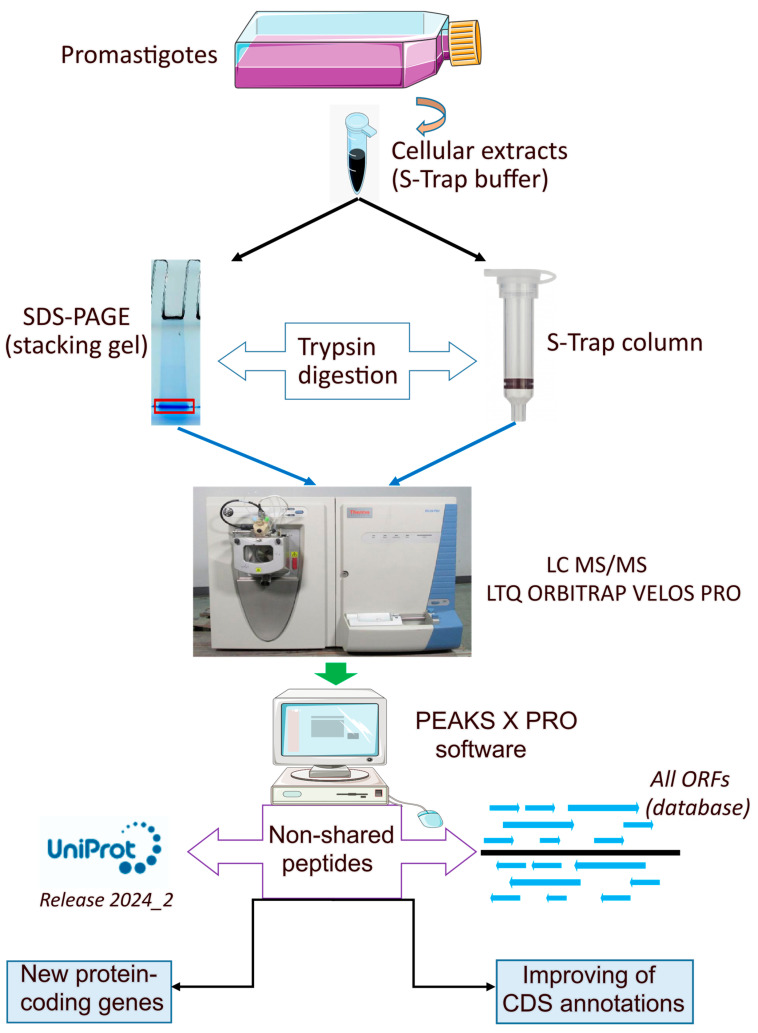
Overview of the experimental and bioinformatics procedures aimed at the identification of new protein-coding genes and improving CDS annotations. Protein extracts derived from *L. donovani* promastigotes were enzymatically digested either in gel (the digested material is shown inside the red square) or in an S-trap column. Afterward, peptide mass spectra were identified by LC–MS/MS using the ion trap LTQ-Orbitrap Velos Pro hybrid mass spectrometer. Mass spectra were searched against the *L. donovani* proteome currently available at UniProt (www.uniprot.org) or a custom database consisting of all possible ORFs found after reading the genome sequence in its six reading frames (named LdHU3-all-ORFs database). Those peptides found only in the latter database led to the identification of new protein-coding genes and the improvement of previously annotated gene models.

**Figure 3 genes-15-00775-f003:**
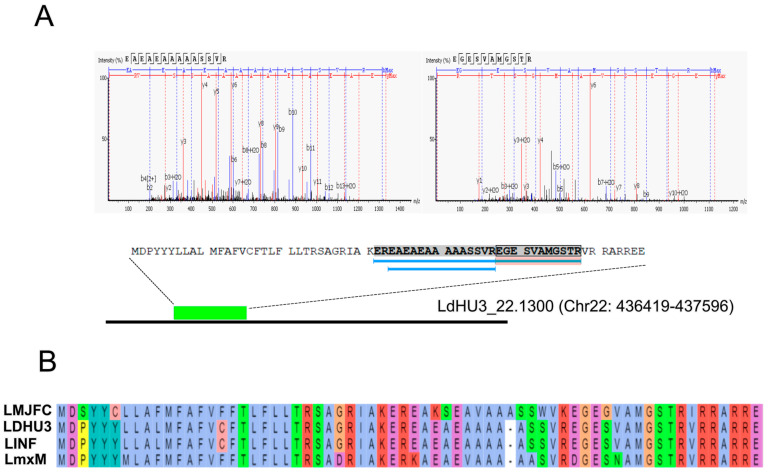
Identification of LDHU3_22.1300 as a new protein-coding gene. (**A**) Mass spectra allowed the identification of three peptides mapping on a theoretical ORF located at transcript LDHU3_22.1300, annotated in the *L. donovani* chromosome 22 as non-coding [22]. (**B**) Multiple alignments between the new protein LDHU3_22.1300 (LDHU3) and those uncovered in the genome of other *Leishmania* species. LMJFC corresponds to an ORF coding for a well-conserved amino acid sequence found in the *L. major* (Friedlin) transcript LMJFC_220016800_t1, which is 867 nucleotides in length and is currently annotated as ncRNA gene (https://tritrypdb.org, accessed on 13 May 2024). Protein LmxM was found in the *L. mexicana* reference genome (MHOM/GT/2001/U1103), in a putative ORF located at chromosome 22 (LmxM.22; coordinates: 389689–389830). The gene coding for the orthologue protein (LINF_220015750; LINF in the figure) was previously identified in *L. infantum* (JPCM5) following also a proteogenomic strategy [16] and its sequence is available as a Mendeley dataset (https://data.mendeley.com/datasets/rrs42p32y9/1, accessed on 13 May 2024). Multiple sequence alignment was carried out by the Clustal Omega tool, and the amino acids were coloured by this tool according to their physicochemical properties [29].

**Figure 4 genes-15-00775-f004:**
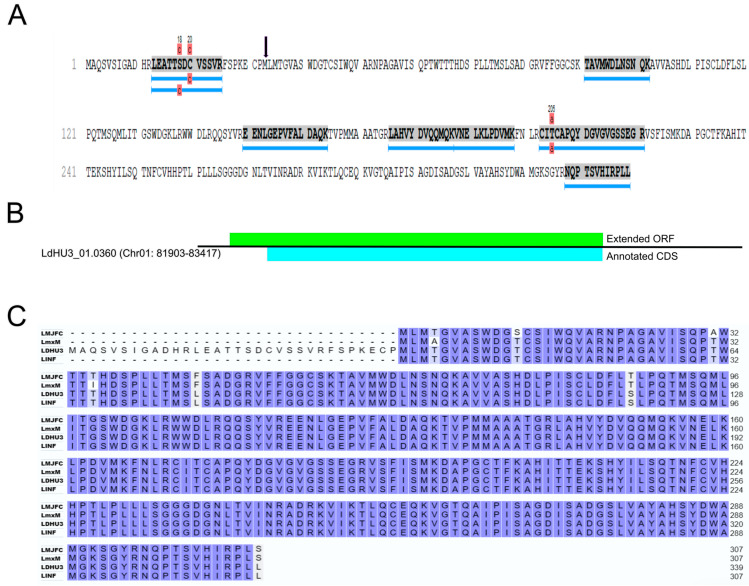
Experimental data leading to improving CDS annotation for gene LDHU3_01.0360. (**A**) Peptides mapped in a theoretical ORF predicted in the *L. donovani* genome sequence, two of them were derived from a region located upstairs of the currently annotated LDHU3_01.0360 CDS (a vertical arrow points to the currently annotated initial methionine). (**B**) Location of the extended ORF (shaded in green) and the currently annotated LDHU3_01.0360 CDS (shaded in blue) on the LDHU3_01.0360 transcript (chromosome coordinates for the transcript are indicated). (**C**) The N-terminal extension experimentally found for the protein encoded by gene LDHU3_01.0360 (LDHU3) is absent in the orthologous proteins currently annotated for other *Leishmania* species: LMJFC_010008400 in *L. major* Friedlin (LMJFC), LmxM.01.0320 in *L. mexicana* U1103 (LmxM) and LINF_010008200 in *L. infantum* JPCM5 (LINF). When the amino acids are identical in all proteins are shaded in dark blue, or they are shaded in light blue when 3 out of 4 are identical.

**Figure 5 genes-15-00775-f005:**
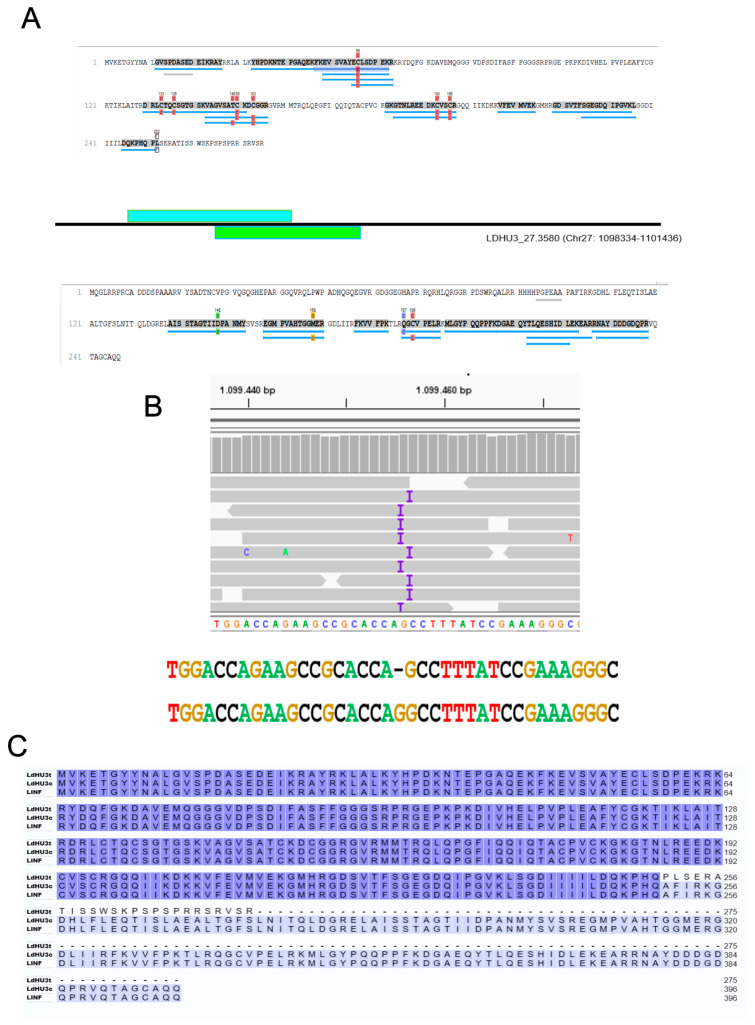
Insertion of a nucleotide in the CDS of gene LdHU3_27.3580 corrected the annotated protein sequence. (**A**) Upper panel shows the identified peptides mapping on the currently annotated protein LdHU3_27.3580 (derived from CDS shaded in blue), and the bottom panel shows the peptides translated from a theoretical ORF (shaded in green) mapped also on the same LdHU3_27.3580 transcript (genomic coordinates are indicated). (**B**) A nucleotide insertion missing in the *L. donovani* genome sequence was observed after mapping the Illumina reads generated by sequencing the genomic DNA of this species. The current assembled sequence and the corrected one are shown at the bottom, respectively. (**C**) Multiple alignments of the current annotated sequence for protein LdHU3_27.3580 (LdHU3t), the amino acid sequence after insertion of G at position 1.099.456 in the *L. donovani* chromosome 27 (LdHU3c) and the orthologous protein LINF_270032200 (LINF). Image in panel B was created using the IGV.2_14.0 tool [31]. When the amino acids are identical in all proteins are shaded in dark blue, or they are shaded in light blue when 2 out of 3 are identical.

**Table 1 genes-15-00775-t001:** New protein-coding genes annotated in this study.

Gene ID	Mass (Da)	#Peptides	#Unique	Product
LDHU3_02.0870	17,391	3	1	Peptidase M3A/M3B family member
LDHU3_05.1170	50,820	5	5	Protein of unknown function
LDHU3_08.0490	124,532	4	4	Protein of unknown function
LDHU3_11.1460	208,953	3	3	ATP-binding cassette subfamily A, member 1
LDHU3_11.1500	208,645	3	3	ATP-binding cassette protein subfamily A, member 4
LDHU3_11.1540	208,645	3	3	ATP-binding cassette protein subfamily A, member 4
LDHU3_20.1640	17,036	6	6	Small myristoylated protein 4
LDHU3_22.1300	7394	3	3	Protein of unknown function
LDHU3_27.0640	654,271	62	59	Calpain-like cysteine peptidase
LDHU3_29.3160	69,485	7	7	Domain of unknown function (DUF4139)
LDHU3_29.3180	195,556	2	2	UDP-glucose/Glycoprotein Glucosyltransferase
LDHU3_30.5010	11,969	2	2	Protein of unknown function
LDHU3_32.4380	58,558	21	21	T-complex protein 1 subunit α|TCP1α|CCT-alfa
LDHU3_32.4600	9779	2	2	Protein of unknown function
LDHU3_33.4490	132,996	6	6	Protein of unknown function
LDHU3_34.1180	185,525	6	2	Flagellar attachment zone protein
LDHU3_34.1190	281,656	9	5	Flagellar attachment zone protein|FAZ1
LDHU3_35.0470	43,405	12	11	ATP-dependent DEAD-box RNA helicase|DHH1
LDHU3_35.6550	90,643	6	6	Zinc finger protein family member|ZC3H28
LDHU3_36.7950	5153	3	3	Protein of unknown function

**Table 2 genes-15-00775-t002:** Categories overrepresented among the proteins having methylated residues.

Functional Category	Proteins *
Ribosomal proteins	eIF4A1, uL16, eL8, eL40, EF1G, eS21, uS8, eS12, eS4, uL1, uS15, eS6, uS19, uL11, uL29, eS26, uS11, RACK1, eL13, eL40, uL3, eEF1Bβ, uL3, eS1, eEF2, uS13, eS10, eEF1Bα, L10a
Protein folding	HOP, Aha1, HSP100, HSP110, HSP70.4, CCT-β, GRP78, HSP70, mtHSP70, HSP83/90, TRAP-1, HOP2, HSP60, Cyp19
RNA-binding proteins	TSR1, SNU13, RBP42, HEL67, DRBD18, ALBA3, DRBD2, RNA helicase, PABP2, PUF11, ribonucleoprotein p18, L-PSP
Oxidative stress	Thioredoxin, tryparedoxin peroxidase, glutathione peroxidase-like protein, tryparedoxin 1 (TXN1), iron superoxide dismutase
Flagellar proteins	PFR2, PFR1, KHAP1, flagellum targeting protein kharon1
Proteases	Aminopeptidase, carboxypeptidase CP1, calpain-like cysteine peptidase

* See Appendix A for retrieving the IDs of the corresponding gene/proteins.

## Data Availability

The data presented in this study are openly available in the Proteo-meXchange Consortium via the PRIDE partner repository at the dataset identifier PXD051920 and DOI: 10.6019/PXD051920.

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
