# Peer review of "A Proteogenomic Approach to Unravel New Proteins Encoded in the *Leishmania donovani* (HU3) Genome"

_genes, 2024, doi:10.3390/genes15060775_

Round 1

Reviewer 1 Report

Comments and Suggestions for Authors

The authors in their study "A proteogenomic approach to unravel new proteins encoded in the Leishmania donovani (HU3) genome" utilized high-throughput proteomics data generated by increasingly more sensible mass spectrometers to recognize and annotate novel proteins from Leishmania donovani (HU3 strain)

The study updates our knowledge regarding proteome and interactome of Leishmania donovani, however aside from the high throughput data, the study did not offer clinical value in the level of therapeutics or diagnostics.

The study should provide possible links of the identified proteins and pathogenesis or novel therapeutics.

Comments on the Quality of English Language

Moderate editing of English language required

Reviewer 2 Report

Comments and Suggestions for Authors

The objectives of the study must be clearly defined and described.

The authors must also explain in detail the gaps in the literature that are covered by this particular study.

Also, the authors must comment on the advantages of their approach and methodologies followed over those of previous similar studies performed previously.

Control strain. Please describe the positive and negative control Leishmania strains included in the present study.

For all the assays employed in the study, please describe the positive and negative controls used in the studies.

The authors are commended for making public their findings and for depositing the spectrometry data in a public repository.

For all sections of the results: please decrease the length of the text, as this is boring and difficult for readers and please include many tables with presentation of the results.

Tables. The lack of tables decreases the value of the manuscript. Please see previous comment.

Figures. The figures should be presented in larger size.

Please include Discussion as a separate section.

References. These are ok.

Conclusions. The conclusions do not fully abide with the results of the study. The conclusions should be rewritten to be better in line with the findings of the study.

Overall. Extensive corrections are necessary before possible acceptance. The manuscript should be improved and should be re-reviewed after revision.

Round 2

Reviewer 1 Report

Comments and Suggestions for Authors

I would accept the paper for publication after minor editing of the language and avoid self citations

Comments on the Quality of English Language

None

Author Response

We have done a careful reading of the manuscript, trying to avoid grammatical and semantic errors. The modifications introduced in the revised manuscript are shadowed in green on the Word file.

However, we cannot eliminate indiscriminately all the articles because, among other reasons, the study reported in this manuscript is a continuation of previous studies. We can eliminate some self-citations if the reviewer gives us reasonable arguments.

Reviewer 2 Report

Comments and Suggestions for Authors

The authors have resolved all the issues and have improved the manuscript.

Author Response

We are glad you agree with how we addressed your initial comments and suggestions. Thank you for the time invested in reviewing our manuscript and for your advice.